# Bioinspired Polyvinyl Alcohol-Based Foam Fabricated via Supercritical Carbon Dioxide Foaming for Atmospheric Water Harvesting

**DOI:** 10.3390/biomimetics10090599

**Published:** 2025-09-08

**Authors:** Yingying Chen, Changjun Guo, Hao Wang, Jiabao Lu, Heng Xie, Ting Wu

**Affiliations:** 1Key Laboratory of Material Chemistry for Energy Conversion and Storage, Ministry of Education; Hubei Key Laboratory of Material Chemistry and Service Failure and Hubei Engineering Research Center for Biomaterials and Medical Protective Materials, Huazhong University of Science and Technology, Wuhan 430074, China; 2School of Materials Science and Engineering, Wuhan Institute of Technology, Wuhan 430205, China

**Keywords:** atmospheric water harvesting, polyvinyl alcohol, supercritical carbon dioxide foaming, photothermal performance, porous foam

## Abstract

The intensifying freshwater crisis underscores the critical need for all-weather, low-energy atmospheric water harvesting technologies. Inspired by the scale-like protrusions and interconnected channels of Tillandsia leaves that enable efficient water capture and release, a polyvinyl alcohol-based foam featuring a three-dimensional porous structure is fabricated using the supercritical carbon dioxide foaming technology. Compared to the traditional freeze-drying method, this approach significantly reduces preparation energy consumption and shortens the production cycle. Lithium chloride integration endows the foam with exceptional moisture absorption capacity, reaching 300% of its weight. Leveraging graphene’s outstanding photothermal conversion properties, the foam achieves a photothermal dehydration rate of 80.7% within 80 min under 1 Sun irradiation, demonstrating a rapid water release capacity. Furthermore, the polyvinyl alcohol-based foam exhibits no performance degradation after 60 cycles, indicating remarkable stability. This technology provides a scalable, low-cost, and all-climate-applicable solution for water-scarce regions.

## 1. Introduction

Freshwater resources, as fundamental resources supporting human survival and social development, are facing an increasingly severe shortage due to accelerated global industrialization, continuous population growth, and deteriorating freshwater pollution. This has become one of the core challenges restricting global sustainable development [1,2,3]. Atmospheric water harvesting (AWH) technology, leveraging the key advantage of “directly capturing water from the air,” breaks through the limitations of climate, geography, and infrastructure, emerging as a groundbreaking solution for water supply in water-scarce areas [4,5,6,7,8]. To date, researchers have developed various high-efficiency AWH materials [9,10,11,12,13], including metal–organic frameworks (MOFs) [14], covalent organic frameworks (COFs) [15], and hygroscopic salt-loaded hydrogels [16]. However, existing materials have limitations: although MOFs and COFs exhibit excellent adsorption performance, their high raw material costs hinder large-scale production; hygroscopic salt-loaded hydrogels, while low-cost, primarily rely on freeze-drying for preparation. This method has inherent drawbacks: it requires maintaining temperature below −20 °C, resulting in extremely high energy consumption, and its long preparation cycle leads to low production efficiency, limiting the large-scale application of hygroscopic salt-loaded hydrogels [17,18,19]. The emerging supercritical carbon dioxide (scCO_2_) foaming technology provides a new approach to address the above preparation challenges [20,21,22,23,24]. Carbon dioxide is the most commonly used physical blowing agent in the preparation of microcellular foamed materials due to its advantages of low cost, non-toxicity, non-flammability, and mild critical conditions (Tc = 31.05 °C, Pc = 7.375 MPa) [25]. ScCO_2_ foaming technology offers advantages such as short preparation cycles and high production efficiency; meanwhile, the uniform microporous structure of supercritically foamed materials can effectively disperse stress, maintaining initial performance after multiple cycles [26,27,28,29].

To address the limitations of existing AWH materials, including high cost of MOFs/COFs and high energy consumption of freeze-dried hydrogels, and leverage the advantages of scCO_2_ foaming, this work aims to develop a bioinspired polyvinyl alcohol-based foam (PVAF) for AWH. Guided by the water capture mechanism of Tillandsia leaves, LiCl is integrated for enhanced moisture absorption and graphene for photothermal dehydration, with the goal of achieving a low—cost, scalable material featuring high moisture absorption efficiency, rapid water release, and excellent cyclic stability.

## 2. Experimental Section

### 2.1. Materials

Polyvinyl alcohol (PVA-117) was purchased from Kuraray (Shanghai, China) Co., Ltd. Anhydrous lithium chloride (LiCl, AR) particles were supplied by Chengdu Kelong Chemical (Chengdu, China) Co., Ltd. Glutaraldehyde (GD, 50%) was obtained from Shanghai Macklin Biochemical Technology (Shanghai, China) Co., Ltd. Graphene (purity ≥ 96%, average diameter < 10 μm) was purchased from Changhongda (Shenzhen, China) Technology Co., Ltd. All chemicals were used as received without further purification. Deionized water was used in this work.

### 2.2. Sample Preparation

The preparation of PVAF is shown in Figure 1. The first step was to prepare PVA/LiCl premixed composite melt via melt blending. First, 0.4 g of glutaraldehyde (50%) solution and 28 g of deionized water were weighed and mixed uniformly. Glutaraldehyde was used as a cross-linking agent for PVA because its aldehyde groups can form covalent hemiacetal/acetal bonds with the hydroxyl groups of PVA which can enhance the structural stability of the PVA matrix. The mixture was blended with 40 g of PVA particles, stirred evenly, and sealed for standing to allow PVA particles to absorb water and fully swell. During this period, different masses of LiCl particles (16, 20, 24, 28 g) were weighed. After PVA particles were fully swollen, the weighed LiCl particles were added and stirred rapidly to mix swollen PVA and LiCl uniformly. The homogeneous mixture was placed in a stretch-dominated internal mixer for melt blending at 100 °C to fully fuse the components, preparing irregular PVA/LiCl premixed composite melt. The second step was to prepare PVA/LiCl/GN premixed flakes: the premixed PVA/LiCl composite melt was molded into 0.15 cm-thick flakes using a plate vulcanizer, then the graphene was sprayed on its surface. The third step was to prepare PVAF via scCO_2_ foaming: the PVA/LiCl/GN premixed flakes were cut into 2 × 2 × 0.15 cm^3^ samples, subjected to moisture absorption in a constant temperature and humidity chamber (25 °C, 98% RH) for 2 h, then these samples were placed in a scCO_2_ foaming kettle for scCO_2_ foaming at constant temperature and pressure.

### 2.3. Material Characterization

Scanning electron microscopy (SEM, SU8010, Hitachi, Japan) was used to characterize the morphology at an accelerating voltage of 5.0 kV. Surface elements were analyzed via energy-dispersive spectroscopy (EDS) combined with SEM. Full-spectrum (300–1100 nm) sunlight was generated using a solar simulator (CELS500L, AULTT, Beijing, China) and monitored using a power meter (CEL-FZ-A, Ceaulight, Beijing, China). Temperature was measured in real-time using an infrared thermal imager (DMI220, Dongmei, Shenzhen, China), and mass changes were recorded using an electronic balance (accuracy = 0.1 mg).

## 3. Results and Discussion

### 3.1. Exploration of Foaming Process

For the scCO_2_ foaming step, the molded PVA mixed melt flakes (40 wt% LiCl) were cut into 2 × 2 × 0.15 cm^3^ samples to explore the foaming conditions. Various factors affect scCO_2_ foaming; first, the effect of foaming temperature on the foaming performance of PVA mixed melt flakes was investigated. The PVA samples were placed in a reactor for scCO_2_ foaming at a pressure of 180 bar, with foaming temperatures set to 120 °C, 110 °C, 100 °C, 90 °C, and 80 °C, respectively. After holding the pressure for 2 h, PVA foamed samples were obtained. The results are shown in Figure 2a: at 120 °C, the PVA sample showed obvious coking without significant cell formation, indicating unsuccessful foaming. At 110 °C, the PVA sample formed obvious cells with slight coking at the edges, indicating partial foaming. At 100 °C, the PVA sample did not fully foam immediately after pressure release but fully foamed after standing for a few minutes without coking. At temperatures below 100 °C, the PVA sample formed obvious cells without coking but failed to fully foam even after standing. This is likely because low temperatures insufficiently softened PVA, hindering the full penetration of scCO_2_ into PVA. Therefore, within the investigated range, the optimal foaming temperature for scCO_2_ foaming of PVA samples was determined to be 100 °C. Next, the effect of foaming pressure on the foaming performance of PVA mixed melt flakes was explored. The PVA samples were foamed at 100 °C with pressures set to 210 bar, 180 bar, 150 bar, 120 bar, and 100 bar, respectively, and held for 2 h. The results are shown in Figure 2b: at pressures above 180 bar, the PVA samples fully foamed but required standing for a few minutes after pressure release. At pressures below 180 bar, the PVA samples failed to fully foam due to insufficient penetration of scCO_2_ caused by low pressure. Thus, within the investigated range, the optimal foaming pressure was selected as 180 bar. After scCO_2_ foaming, the samples fully foamed immediately after pressure release without standing. As shown in Figure 2c, the unabsorbed sample had a foaming ratio of 295% (incomplete foaming), while samples foamed after 2 h and 4 h moisture absorption in a constant temperature and humidity chamber (25 °C, 98% RH) showed foaming ratios of 870% and 890% with complete foaming and no coking. Due to the slight difference in foaming ratios, 2 h moisture absorption (25 °C, 98% RH) in a constant temperature and humidity chamber was chosen as the standard condition.

Finally, the effect of different LiCl contents on the foaming performance of PVA mixed melt flakes was investigated. As shown in Figure 2d, under the conditions of 100 °C, 180 bar, and 2 h moisture absorption, the sample with 40% LiCl showed a foaming ratio of 870%, significantly higher than those with other contents. The foaming ratios for 50%, 60%, and 70% LiCl were 595%, 489%, and 585%, respectively. The decrease may be attributed to the enhanced adsorption and diffusion of water molecules by LiCl at higher concentrations, leading to reduced structural stability due to excessive moisture absorption, weakened support capacity, and further damage to cell stability.

### 3.2. Characterization of PVAF

A digital camera image of the PVAF is shown in Figure 3a–c presenting scanning electron microscopy (SEM) images of foamed samples without and with LiCl, respectively. Both samples exhibit a distinct three-dimensional porous structure with similar pore sizes. The LiCl-free sample has very smooth and relatively uniform cells, while the LiCl-containing sample has rough cells with clearly visible LiCl particles attached to the surface. As shown in Figure 3d,e, the LiCl-containing foamed sample has good geometric uniformity. FTIR was used to confirm the changes in the properties of PVA after adding glutaraldehyde and LiCl. As shown in Figure 3f, the pure PVA spectrum shows a stretching vibration peak of -OH groups around 3260 cm^−1^. However, in the PVA melt with 40% LiCl, the stretching vibration peak of -OH shifts to 3315 cm^−1^, indicating that the addition of the composite plasticizer disrupts the original intra- and intermolecular hydrogen bonds in PVA, and the newly formed hydrogen bonds between the plasticizer and PVA are relatively weak. Thus, the shift in the hydroxyl stretching vibration peak to higher wavenumbers is considered the main reason for the thermoplastic processability of PVA. As shown, with increasing LiCl content, the hydroxyl stretching vibration peak shifts more obviously to higher wavenumbers, indicating that the composite plasticizer can effectively plasticize PVA.

### 3.3. Exploration of Material Adsorption Performance

PVA samples with 0%, 40%, 50%, 60%, and 70% LiCl (relative to PVA mass) underwent moisture absorption in a constant temperature and humidity chamber (25 °C, 98% RH) for 2 h, then subjected to scCO_2_ at 100 °C and 180 bar for 2 h to obtain the PVA samples with different LiCl contents. The PVA samples were placed in a constant temperature and humidity chamber (25 °C, 98% RH) for moisture absorption tests, and mass changes were recorded using a balance. The results are shown in Figure 4a: the moisture absorption effect differs significantly between LiCl-free and LiCl-containing foamed samples. The LiCl-free sample has a moisture absorption rate below 80%, showing poor performance. Benefiting from the three-dimensional structure and hydrophilicity of PVA, as well as the strong hygroscopicity of LiCl, samples with 40%, 50%, 60%, and 70% LiCl exhibit excellent water capture and storage capabilities. After moisture absorption, the samples can effectively confine water within the three-dimensional structure, with moisture absorption efficiencies of 86.9%, 267.2%, 301.4%, 307.5%, and 319.8%, respectively, outperforming many reported materials (Figure 4b) [30,31,32,33,34,35,36,37,38]. Among them, samples with 50%, 60%, and 70% LiCl achieve moisture absorption efficiencies above 300% of their weight. Due to the slight differences in efficiency and considering cost, the 50% LiCl sample was selected for subsequent experiments. To further elucidate the excellent adsorption performance, a diagram was drawn to describe the moisture absorption mechanism of the PVA sample. As shown in Figure 4c, the PVA sample has a large specific surface area and extraordinary moisture absorption capacity, enabling strong surface adsorption of water molecules. When the dry sample is exposed to a humid environment, the presence of LiCl reduces the saturated vapor pressure at the interface between the sample and air, guiding water to adhere to the outer surface.

Specifically, LiCl distributed on the periphery of the polyvinyl alcohol matrix skeleton absorbs water first. Additionally, the molecular structure of PVA contains numerous hydrophilic hydroxyl groups, which can form hydrogen bonds with water molecules, facilitating water absorption. As more water accumulates, LiCl in the peripheral region of the three-dimensional network begins to dissociate and form hydrated ions, reducing salt concentration. Coupled with capillary action, water molecules diffuse toward the inner region of the three-dimensional network with higher salt concentration. As water diffuses inward, the peripheral water decreases, allowing the sample to continue absorbing water. This cycle continues until LiCl is fully hydrated and dissociated, and polymer chains unfold, macroscopically manifesting as sample volume expansion.

### 3.4. Exploration of Material Surface Temperature and Desorption Performance

To achieve recyclable moisture absorption, 0 mg, 1 mg, 2 mg, 3 mg, and 4 mg of photothermal material graphene were sprayed on the surface of PVA/LiCl premixed flakes, labeled as PG0 (0 mg graphene), PV1 (1 mg graphene, hereinafter referred to as PVAF), PG2 (2 mg graphene), PG3 (3 mg graphene), and PG4 (4 mg graphene), respectively. An infrared thermal imager was used to record the surface temperature changes in PVA/LiCl premixed flakes with different graphene contents under xenon lamp irradiation. The results are shown in Figure 5a: the initial temperature was room temperature (~26.1 °C). After 300 s of irradiation at 1 kW·m^−2^, PG0 showed a very slow temperature rise, reaching only 36 °C. This indicates that pure PVA/LiCl material has low photothermal conversion efficiency, making it difficult to achieve rapid temperature rise via irradiation, which directly restricts water desorption efficiency and hinders efficient recyclable moisture absorption. PVAF, PG2, PG3, and PG4 with graphene all exhibited excellent heating performance: within 300 s of irradiation, their surface temperatures exceeded 55 °C with higher heating rates than PG0, indicating that graphene, as a photothermal material, can effectively enhance the photothermal conversion efficiency of PVA/LiCl-based materials, enabling rapid temperature rise under irradiation to provide sufficient energy for water desorption and ensure efficient recyclable moisture absorption. Due to the similar temperature change trends of PVAF, PG2, PG3, and PG4, PVAF was selected for subsequent experiments considering cost while meeting basic performance requirements.

The infrared thermal image of PVAF under one sun irradiation is shown in Figure 5b: the initial temperature was ~26.1 °C; after 2 min of irradiation, the surface temperature of PVAF reached 50.1 °C; after 6 min, 56.1 °C; and after 10 min, 57.3 °C. To further verify the excellent photothermal conversion performance of graphene-sprayed materials, the desorption performance of PG0 and PVAF was tested (Figure 5c). PG0 desorbed only 60% of adsorbed water within 80 min under one sun irradiation, while PVAF desorbed 80.7% within the same time, significantly outperforming PG0. This is attributed to the good photothermal performance of graphene, enabling rapid and smooth dehydration of PVAF. To further elucidate the excellent dehydration performance, a diagram was drawn to describe the dehydration mechanism of PVAF. As shown in Figure 5d, under irradiation, the sample first removes free water containing dissolved LiCl, and LiCl gradually precipitates. At this stage, water stored in the PVA network remains unremoved, and the network remains swollen. With prolonged irradiation, some LiCl precipitates. Due to concentration difference and capillary action, water gradually transfers to the light-exposed surface of the sample. Meanwhile, water stored in the PVA network is gradually desorbed with extended irradiation time. The reduction in internal water strengthens the hydrogen bonds between water molecules and the interactions between water and polymer chains. As a porous elastomer, the PVA network contracts due to these strengthened interactions.

### 3.5. Exploration of Moisture Absorption-Dehydration Cycle Performance

To further demonstrate its excellent performance, cyclic tests were conducted on PVAF, including moisture absorption tests and dehydration tests. Specifically, building on previous findings, experiments showed that the 50% LiCl sample, tested for photothermal dehydration using a xenon lamp simulating sunlight with mass changes recorded by a balance, could successfully desorb 80% of absorbed water. Thus, the cyclic moisture absorption and dehydration of the sample were explored using a constant temperature and humidity chamber and a xenon lamp: the sample was dehydrated photothermally after absorbing ~300% moisture, then reabsorbed moisture in the constant temperature and humidity chamber after desorbing ~80% water. This cycle was repeated, and mass changes within five cycles were recorded, with no water overflow on the sample surface during the process.

As shown in Figure 6a, within the five tested cycles, the sample absorbed over 300% of its weight in moisture and desorbed at least 80% water. For further exploration of long-term cyclic performance, high-saturation humidity and high light intensity were used to shorten the total cycle time and verify the durability of moisture absorption and dehydration. As shown in Figure 6b, since the sample was dehydrated after absorbing ~300% moisture (desorbing at least 80% water), no water overflow (i.e., no LiCl loss) occurred. To date, the prepared PVAF has maintained excellent cyclic moisture absorption and dehydration performance after 60 cycles. EDS tests were performed on the sample surface before moisture absorption, after 5 cycles, and after 60 cycles (Figure 6c–e). The EDS spectra show that the distribution of the Cl element in the sample is relatively uniform before and after moisture absorption, with little change in mass ratio. During moisture absorption, the interaction between salt ions and water molecules is more significant than that between water molecules. According to the concentration difference law, the internal concentration difference in the sample during moisture absorption generates a driving force for water diffusion, causing water to penetrate from the exterior to the interior where the LiCl content is higher. With sufficient moisture absorption time, the LiCl concentration in the entire sample tends to reach a balance. From the perspective of molecular thermal motion, all natural processes proceed toward increasing disorder of molecular thermal motion. A concentration difference inside the sample can be regarded as an ordered state, generating a driving force toward disorder, i.e., osmotic pressure. The driving force decreases as the internal and external concentration difference narrows, until disorder is achieved. If LiCl moves with water molecules, a concentration difference would persist, violating the law of molecular thermal motion. On the other hand, during melt blending, LiCl is uniformly embedded in the PVA matrix, which exerts a confinement effect on LiCl. Combined with the EDS results in Figure 6c–e, the Cl element distribution on the sample surface remains uniform after 5 and 60 cycles, with no significant change in mass ratio. EDS was further used to analyze the distribution of LiCl during moisture absorption and dehydration: Figure 6f–h show EDS results of the surface, interior, and bottom layer of the sample after 60 cycles. The Cl element distribution is relatively uniform across the surface, interior, and bottom layer with no obvious change in mass ratio, further confirming the confinement effect of the PVA matrix on LiCl, which effectively prevents LiCl aggregation and ensures moisture absorption efficiency. In summary, Li^+^ ions do not aggregate significantly at the bottom during moisture absorption. During dehydration, water escapes from the upper surface as water vapor. As water decreases, the LiCl concentration on the upper surface becomes higher than elsewhere; similarly, according to the concentration difference law, water is transported to the upper surface, with the diffusion driving force mainly transporting water. Thus, Li^+^ ions do not aggregate on the upper surface during dehydration.

## 4. Conclusions

Inspired by the atmospheric water capture of Tillandsia leaves, a PVAF with excellent cyclic stability and a short preparation cycle is prepared using air spraying combined with scCO_2_ foaming technology. PVAF has a three-dimensional self-supporting porous structure with uniformly embedded LiCl. Benefiting from the strong hygroscopicity of LiCl, PVAF exhibits an excellent moisture absorption efficiency of 300% of its weight. Graphene endows PVAF with good photothermal dehydration performance, achieving a dehydration efficiency of 80.7% within 80 min and maintaining performance after 60 cycles. ScCO_2_ foaming technology significantly shortens the preparation cycle and reduces energy consumption, making it suitable for large-scale production. Meanwhile, PVAF exhibits excellent performance stability, thus enabling all-weather operation, which involves solar-driven water production during the day and passive moisture absorption at night, for cyclic atmospheric water harvesting. PVAF is expected to provide an economical and sustainable distributed freshwater supply for arid and remote regions worldwide.

## Figures and Tables

**Figure 1 biomimetics-10-00599-f001:**
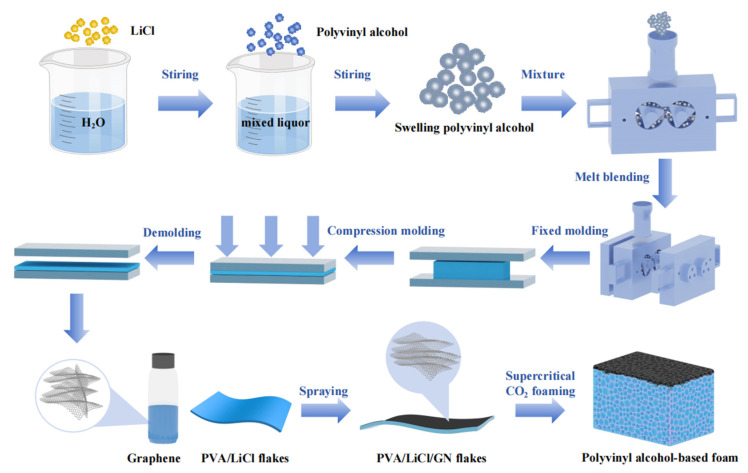
Preparation process of PVAF.

**Figure 2 biomimetics-10-00599-f002:**
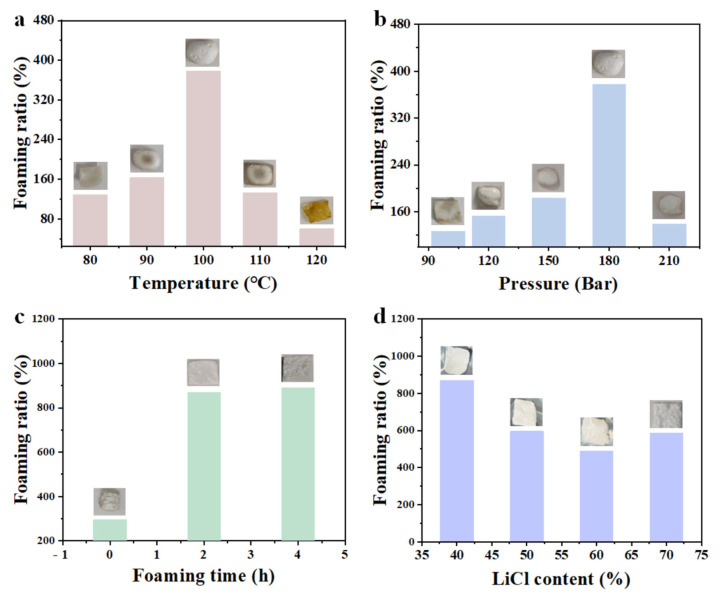
(**a**) Foaming ratios of PVA mixed melt flakes at different foaming temperatures; (**b**) Foaming ratios of PVA mixed melt flakes at different foaming pressures; (**c**) Foaming ratios of PVA mixed melt flakes with different moisture absorption times; (**d**) Foaming ratios of PVA mixed melt flakes with different lithium chloride contents.

**Figure 3 biomimetics-10-00599-f003:**
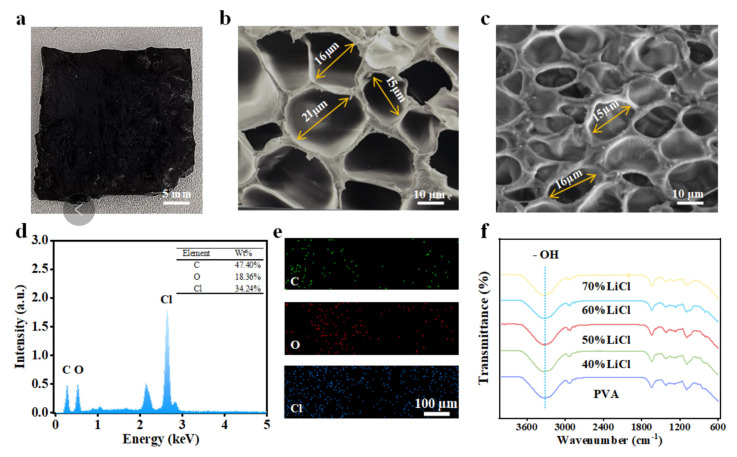
(**a**) Digital camera image of PVAF; (**b**) SEM image of foamed sample without lithium chloride; (**c**) SEM image of foamed sample with lithium chloride. (**d**) EDS image of foamed sample with lithium chloride; (**e**) Distribution of C, O, and Cl elements in the foamed sample with lithium chloride; (**f**) FTIR spectra of PVA melts with different lithium chloride contents.

**Figure 4 biomimetics-10-00599-f004:**
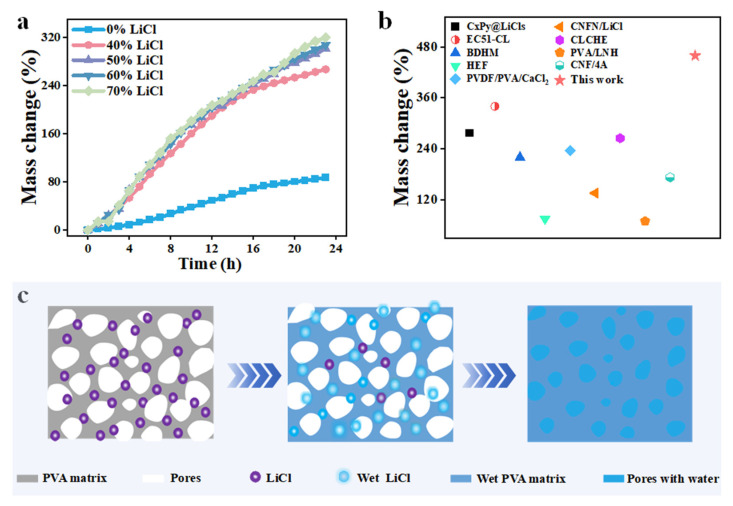
(**a**) Moisture absorption curves of samples with different lithium chloride contents; (**b**) Comparison of moisture absorption performance between the PVA samples and other studies [30,31,32,33,34,35,36,37,38]; (**c**) Schematic diagram of moisture absorption of PVA sample.

**Figure 5 biomimetics-10-00599-f005:**
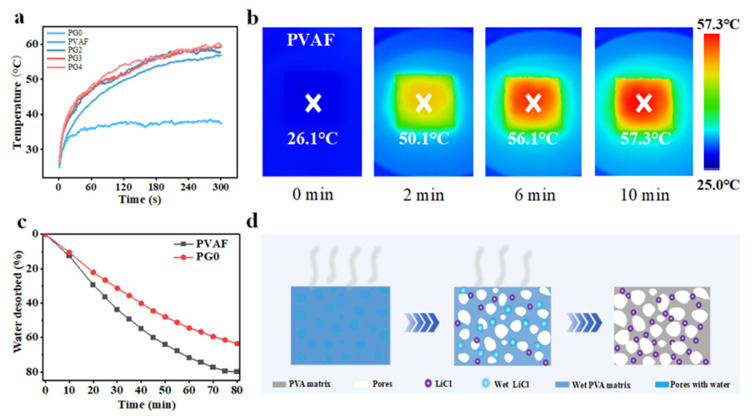
(**a**) Surface temperature change curves of PVA/LiCl premixed flakes with different graphene contents under 1 kW·m^−2^ irradiation; (**b**) Infrared thermal image of PVAF under 1 kW·m^−2^ irradiation; (**c**) Dehydration curves of PG0 and PVAF. (**d**) Schematic diagram of the dehydration of PVAF.

**Figure 6 biomimetics-10-00599-f006:**
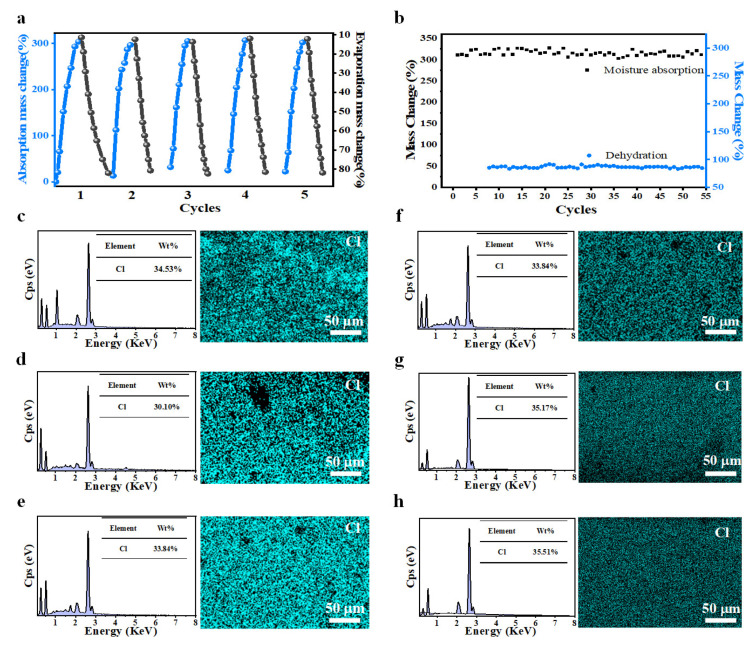
(**a**) Moisture absorption and dehydration curves of 50% LiCl sample within 5 cycles; (**b**) Moisture absorption and dehydration of 50% LiCl sample within 60 cycles; (**c**–**e**) EDS spectrum and Cl element distribution of 50% LiCl sample before moisture absorption, after 5 cycles, and after 60 cycles; (**f**–**h**) EDS spectrum and Cl element distribution of the surface, interior, and bottom layer of the sample after 60 cycles.

## Data Availability

The original contributions presented in this study are included in the article. Further inquiries can be directed to the corresponding authors.

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
