# Peer review of "Bioinspired Polyvinyl Alcohol-Based Foam Fabricated via Supercritical Carbon Dioxide Foaming for Atmospheric Water Harvesting"

_biomimetics, 2025, doi:10.3390/biomimetics10090599_

Round 1

Reviewer 1 Report

Comments and Suggestions for Authors

This manuscript addresses the pressing freshwater crisis by developing an biomimetic atmospheric water harvesting material through supercritical carbon dioxide (scCO₂) foaming technology. Inspired by the natural water-capturing ability of Tillandsia leaves, the research fabricates a polyvinyl alcohol (PVA)-based foam with a three-dimensional porous structure via supercritical carbon dioxide (scCO₂) foaming technology. The integration of lithium chloride (LiCl) for enhanced moisture absorption and graphene for efficient photothermal dehydration, coupled with the demonstrated cycling stability, highlights the study’s innovative value and practical relevance. The core research content of this article is complete, the academic logic is clear, and the format and language expression have basically conformed to the norms. Howerver, some minor details can be further optimized to enhance the reading experience and completeness of the manuscript. The specific suggestions are as follows:

  1. When "supercritical carbon dioxide foaming technology" was first mentioned, although the abbreviation "scCO₂" was marked, it was not unified in subsequent expressions (some parts still use the full name). It is suggested that the consistency of the abbreviation or the full name be maintained in the manuscript.
  2. It is mentioned in section 3.2 that "the stretching vibration peak of -OH shifts to 3315 cm⁻¹", but in the corresponding infrared spectrum diagram,there is a line highlighting the wavenumble of the characteristic peak of the hydroxyl group as "3260 cm⁻¹". It is suggested that it is more reasonable to mark the line as functional group "-OH" on the chart.
  3. Some figures suffer from insufficient clarity, eg. Figure 6c. Check all images for compression distortion (such as color block faults and blurred text) and ensure that the image formats meet the requirements of the target journal.
  4. Some references in the reference list lack page number informationeg, eg. Reference 2, 8, 16. Supplement the complete page number range for journal articlest to ensure readers can accurately locate the original text.
  5. The first line of some paragraphs in the main text should be indented by 2 characters, while some paragraphs should not be indented. It is recommended that the entire text adopt one uniform format, conforming to the layout habits of academic papers.

Author Response

Dear Editors and Referees:

Thanks for your comments on our manuscript entitled “Bioinspired polyvinyl alcohol-based foam fabricated via supercritical carbon dioxide foaming for atmospheric water harvesting (biomimetics-3837758). We sincerely appreciate the valuable comments from the editor and reviewers, which greatly assist us in improving the quality of the manuscript. We have carefully revised the manuscript according to these comments, and point-by-point responses to questions from the editor and reviewer have been given as follows. Detailed explanations/descriptions of the commented areas have been added where necessary and appropriate in the revised manuscript. All the revisions in the revised manuscript have been marked in red.

Responses to the Referees’ comments:

Reviewer #1

Comments 1: When "supercritical carbon dioxide foaming technology" was first mentioned, although the abbreviation "scCO2" was marked, it was not unified in subsequent expressions (some parts still use the full name). It is suggested that the consistency of the abbreviation or the full name be maintained in the manuscript.

Response 1: Thank you for pointing this out. Subsequent expressions have been unified as scCO₂. “Supercritical carbon dioxide foaming technology” in line 53 and line 314 has been changed to “ScCO₂ foaming technology”.

 Comments 2: It is mentioned in section 3.2 that "the stretching vibration peak of -OH shifts to 3315 cm-1", but in the corresponding infrared spectrum diagram, there is a line highlighting the wavenumble of the characteristic peak of the hydroxyl group as "3260 cm-1". It is suggested that it is more reasonable to mark the line as functional group "-OH" on the chart.

Response 2: Thank you for pointing this out. The functional group "-OH" has been marked on the infrared spectrum diagram instead of the wavenumber. The "3260 cm⁻¹" label on the FTIR spectrum in Figure 3f has been replaced with "-OH".

Comments 3: Some figures suffer from insufficient clarity, eg Figure 6c. Check all images for compression distortion (such as color block faults and blurred text) and ensure that the image formats meet the requirements of the target journal.

Response 3: Thank you for pointing this out. Low-clarity images have been reprocessed, and the format of all images has been verified to meet the journal’s requirements. For Figure 6c, the original compressed image has been replaced with a high-resolution version without color block faults or blurred text, ensuring clear display of element distribution and EDS energy peak information.

Comments 4: Some references in the reference list lack page number information, eg. Reference 2,8,16. Supplement the complete page number range for journal articles to ensure readers can accurately locate the original text.

Response 4: Thank you for pointing this out. The complete page number ranges for References 2, 8, 16, 21, 25, 32 and 37 have been supplemented to ensure readers can locate the original texts accurately.

Comments 5: The first line of some paragraphs in the main text should be indented by 2 characters, while some paragraphs should not be indented. It is recommended that the entire text adopt one uniform format, conforming to the layout habits of academic papers.

Response 5: Thank you for pointing this out. The paragraph indentation format of the entire manuscript has been unified to conform to academic paper layout conventions. All body paragraphs have been unified to "first line indented by 2 characters", the non-indented format of 3 paragraphs has been deleted, and consistent indentation across the entire text has been ensured.

Reviewer 2 Report

Comments and Suggestions for Authors

Dear authors,

I read with maximum attention and interest your paper and I want to congratulate you for your work. The paper is well written, the methods and experiments are clearly explained and the results are concluding.

These are my observations:

  • In paragraph “I. Introduction”, line 61-68 seems to fit better in the conclusion section. Please rephrase this paragraph as a statement of what you want to accomplish, rather than a conclusion to your paper.
  • How Tillandsia leaves inspired your work? Please add a phrase to explain how this plant leads to this work or how these leaves capture and release water.
  • What is the role of glutaraldehyde? Please explain in text why you used this reactant.
  • Paragraph 3.4 The PVAF sample acronym is confusing (I suppose is PVA with 1mg of graphene). Please rename this acronym to PV1 or add an explanation in the text like ”PV1 hereinafter referred to as PVAF”.

Author Response

Dear Editors and Referees:

Thanks for your comments on our manuscript entitled “Bioinspired polyvinyl alcohol-based foam fabricated via supercritical carbon dioxide foaming for atmospheric water harvesting (biomimetics-3837758). We sincerely appreciate the valuable comments from the editor and reviewers, which greatly assist us in improving the quality of the manuscript. We have carefully revised the manuscript according to these comments, and point-by-point responses to questions from the editor and reviewer have been given as follows. Detailed explanations/descriptions of the commented areas have been added where necessary and appropriate in the revised manuscript. All the revisions in the revised manuscript have been marked in red.

Reviewer #1

Comments 1: In paragraph “l. Introduction", line 61-68 seems to fit better in the conclusion section. Please rephrase this paragraph as a statement of what you want to accomplish, rather than a conclusion to your paper.

Response 1: Thank you for pointing this out. Lines 61-68 of Section 1 "Introduction" have been rephrased from a conclusion-style statement to a research objective statement. The original text "Based on the above research background, inspired by the atmospheric water capture of Tillandsia leaves, a polyvinyl alcohol-based foam (PVAF) with a three-dimensional porous network and uniformly embedded LiCl in the matrix is prepared using the scCO₂ foaming combined with air spraying technology... Thus, this work provides a high-performance and economical solution for atmospheric water harvesting." has been replaced with lines 57-63: "To address the limitations of existing AWH materials, including high cost of MOFs/COFs and high energy consumption of freeze-dried hydrogels, and leverage the advantages of scCO₂ foaming, this work aims to develop a bioinspired polyvinyl alcohol-based foam (PVAF) for AWH. Guided by the water capture mechanism of Tillandsia leaves, LiCl is integrated for enhanced moisture absorption and graphene for photothermal dehydration, with the goal of achieving a low-cost, scalable material featuring high moisture absorption efficiency, rapid water release, and excellent cyclic stability."

Comments 2: How Tillandsia leaves inspired your work? Please add a phrase to explain how this plant leads to this work or how these leaves capture and release water.

Response 2: Thank you for pointing this out. An explanation of how Tillandsia leaves inspired the work has been added in Section 1 "Introduction". The original text "inspired by the natural atmospheric water capture capability of Tillandsia leaves" has been unified to lines 15-16: "Inspired by the scale-like protrusions and interconnected channels of Tillandsia leaves that enable efficient water capture and release".

Comments 3: What is the role of glutaraldehyde? Please explain in text why you used this reactant.

Response 3: Thank you for pointing this out. An explanation of the role of glutaraldehyde and the reason for its use has been added in Section 2.2 "Sample Preparation". The lines 75-78 "Glutaraldehyde was used as a cross-linking agent for PVA because its aldehyde groups can form covalent hemiacetal/acetal bonds with the hydroxyl groups of PVA which can enhance the structural stability of the PVA matrix" have been added.

Comments 4: Paragraph 3.4 The PVAF sample acronym is confusing (I suppose is PVA with 1mg of graphene). Please rename this acronym to PV1 or add an explanation in the text like "PV1 hereinafter referred to as PVAF".

Response 4: Thank you for pointing this out. An explicit explanation of the PVAF acronym has been added in Section 3.4 to eliminate confusion. Lines 208-211 ("To achieve recyclable moisture absorption, 0 mg, 1 mg, 2 mg, 3 mg, and 4 mg of photothermal material graphene were sprayed on the surface of PVA/LiCl premixed flakes, labeled as PG0, PVAF, PG2, PG3, and PG4, respectively.") have been modified to: "To achieve recyclable moisture absorption, 0 mg, 1 mg, 2 mg, 3 mg, and 4 mg of photothermal material graphene were sprayed on the surface of PVA/LiCl premixed flakes, labeled as PG0 (0 mg graphene), PV1 (1 mg graphene, hereinafter referred to as PVAF), PG2 (2 mg graphene), PG3 (3 mg graphene), and PG4 (4 mg graphene), respectively."